# Designing and Evaluating Energy Product-Service Systems for Energy Sector (EPSS) in Liberalized Energy Market: A Case Study in Space Heating Services for Japan Household

**Widha Kusumaningdyah [1],\*** , **Benjamin McLellan [2]** and **Tetsuo Tezuka [2]**

[1] Department of Industrial Engineering, Faculty of Engineering, University of Brawijaya, 65145, MT. Haryono 167, Malang 65145, Indonesia

[2] Department of Socio-Environmental Energy Science, Graduate School of Energy Science, Kyoto University, 606-8501, Yoshidahonmachi, Sakyo Ward, Kyoto City, Kyoto 606-8501, Japan; b-mclellan@energy.kyoto-u.ac.jp (B.M.); tezuka@energy.kyoto-u.ac.jp (T.T.)

\* Correspondence: widhadyah@ub.ac.id; Tel.: +62-341-587710, 587711 (ext. 1283)

**Abstract:** This paper introduces Energy-Product-Service Systems (EPSS) to overcome the issues in the liberalized energy market. Currently, conventional energy services are highly commoditized, thus eroding their competitiveness in a liberalized energy market. In addition, customer benefits from the current system are hard to analyse, because it is difficult for customers to observe the quality of the results of energy consumption. EPSS incorporates energy sector with other related sectors to design and deliver immediate result of energy usage that suit to customer's specific needs is expected to provide better performance than current system. However, EPSS service design and implementation involve high risk and performance uncertainties. Therefore, this study proposes a method to design and to evaluate EPSS and compare its performances with energy product/service performance in current system through Simulation-Based Design (SBD). SBD is used to construct EPSS service considering stakeholders' interest and to evaluate the service performance, by simulating alternative scenarios in order to seek conditions that are expected to fulfil stakeholders' requirements. In the proposed analysis, three service features, that is, service consumption management, operational system design and electricity supply management are introduced and used to develop EPSS alternative design for space-heating service. Afterwards, customer satisfaction and the company's benefit for each service scenarios are simulated and compared with the performance of the current system. In this context, EPSS design that includes operational system design and electricity supply management results in better benefit for all stakeholders.

**Keywords:** Energy Product-Service Systems; PSS; energy; service

## 1. Introduction

The notion of Energy-Product-Service Systems (EPSS) is introduced in this paper to tackle issues in the liberalized energy market. EPSS is a system of products, services and networks that incorporate the energy sector with other related sectors to design and deliver the ultimate results of energy usage to suit the customer's specific needs. Its purpose is to improve business competitiveness and users' benefits in a liberalized energy market. However, there are several problems in EPSS introduction and development in the market. On the one hand, to design a service for EPSS that performs well in the liberalized market is difficult. On the other hand, providing empirical evidence of EPSS performance in comparison with existing electricity services is also challenging. Accordingly, this study proposes a

method to design EPSS service and evaluates their performance relative to conventional electricity services through Simulation-Based Design.

The Japanese Liberalized Energy Market has been found to have two key problems in relation to conventional energy services: (1) Highly commoditized energy services, eroding competitiveness and endangering companies' profit and sustainability, (2) Customer benefit from conventional energy service is not well-acknowledged, because it is difficult for customers to observe the quality of the results of energy consumption. Energy has experienced commoditization, eroding its competitiveness and often leading to profit squeeze for retailers [1]. Unfortunately, competition between retailers that offers energy supply dominates the current liberalized market [2]. Thus, the competition is hard for those that strive for competitive prices of energy supply. In addition to this, there is strong encouragement to improve energy efficiency in consumption that runs contrary to the company's objective to maximize sales volumes. Such conditions can severely harm a firm's profit in the short term and its sustainability in the long term. When it is hard for energy suppliers to maintain profits, the competitiveness of the market is risked, with fewer suppliers interested to enter the market. This situation may drive the market towards monopoly or other events that indicate market failure (e.g., negative externalities, inefficiency of resource allocation, etc.). For these reasons, energy supply on its own does not have a clear advantage in a liberalized market.

Residential electricity demand is fundamentally derived from the demand of energy services, such transportation, space heating and lighting [3,4]. Nonetheless, in current system, the service is consumed as an input to households' activity. In addition to that, electricity supply component is fundamentally intangible. As a result, the benefit of electricity supply services is difficult for residential customers to comprehend. Despite the fact that customers purchase and consume energy, they are not clearly aware of the quality of energy consumption and the efficiency or effectiveness of its conversion to electricity services, as it is difficult for the customer to directly evaluate. For this reason, we consider that conventional retail focusing on the supply of electricity does not provide sufficient benefit for residential customers to differentiate from competitors.

We propose Energy Product-Service Systems (EPSS), which are characterized by shifting tangible resource ownership from customers to producers. Omitting the requirement of appliance ownership allows companies to elaborate various service designs that focus on providing energy service performance for the households. It also allows customers to focus on the service quality that fits to their needs and lifestyle. This is different from the incumbent system, which we refer as Product-Oriented Systems (POS), where households must purchase the appliance and the electricity to operate it and deliver and control the service by themselves to get the expected results.

EPSS seems provide better benefit for household customer and electricity retailers compare to current system. Nonetheless, not all EPSS is better than service under POS by default. Methodical steps to design and evaluated EPSS service performance before implementation becomes indispensable. We proposed Simulation-Based Design to design and evaluate the EPSS performance and compare with the performance of conventional energy service in liberalized energy market. In attempt to mitigate the risk from performance uncertainties, the method is developed by considering stakeholders' properties, interest and behaviour in the market.

## 2. Introduction to EPSS

EPSS is defined as a "system" that incorporates energy, product and operations of dwellings by incorporating its basic functional systems for a household. It aims to create win-win conditions for all the stakeholders that involved in the system, by taking their interest into consideration in constructing EPSS service. Being ingredient of wide range of economic activity, energy system becomes crucial and unavoidably is considered as complex system. Energy sector itself consists of industries whereas each has special feature in technology and operation. In addition to that, the system interaction exists at different level and includes multidisciplinary subjects. EPSS is a system that incorporate

energy, product and service, thus the stakeholders vary depend on the system boundary and purpose of discussion.

In this study, EPSS stakeholders consist of household consumer and service provider. We consider energy retailer as the provider of EPSS service in which competes with conventional retailer that offer energy supply service. Energy retailer not only supplies energy sources but also the product required to deliver particular service performance. The product used to deliver EPSS service can be any appliance, equipment or material to perform unit function to create a desirable condition for the households. The service refers to activities or process performed by the actors (e.g., company or household) to create a desirable output from energy consumption. In EPSS, company performs the service. While in incumbent system, the household most likely to perform the service activities.

## 2.1. The Difference between EPSS and Incumbent System (POS)

The main difference between POS and EPSS is the value proposition or simply stated, the offering. Given the same household needs, business in POS offers energy supply separated from supply of appliance and equipment, hence the product ownership transfers to the customer. Transfer of ownership is followed by transfers of the risks and benefits of the product from provider to customer, including the failure and quality of the service that are created by the customer himself/herself. In contrast, there's no obligation for households to purchase the service resources, because EPSS deliver a result or functional unit of consumption. Thus, product ownership belongs to the service provider. EPSS business model enables the customer to focus on the quality of the service instead of expecting some product quality that might be less or beyond their needs. Eventually, EPSS implementation leads to modification of actors' interactions shown in Figure 1. In consequence, the actors on POS and EPSS behave differently as shown on Figures 2 and 3.

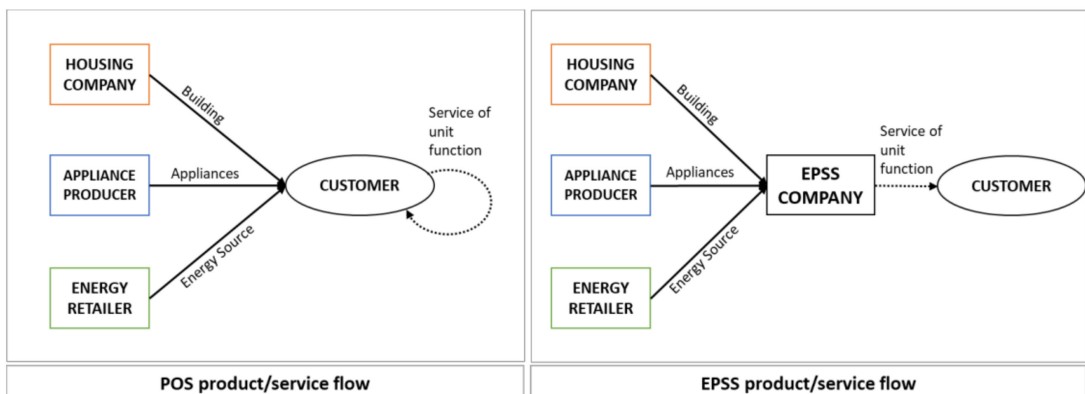

**Figure 1.** Energy, product and service flow in product oriented systems (POS) and energy product service systems (EPSS).

In POS consumers are left to transform the product purchase into something that effectively fulfils their need. The standard way in POS is that company produces goods, trades it with user and receives the payment. On the other hand, consumers are left alone and given freedom to finance the purchase, to learn how to use and extract the benefit from the product. Consumers also more likely required doing maintenance, including purchasing complementary parts or component if needed (see Figure 2).

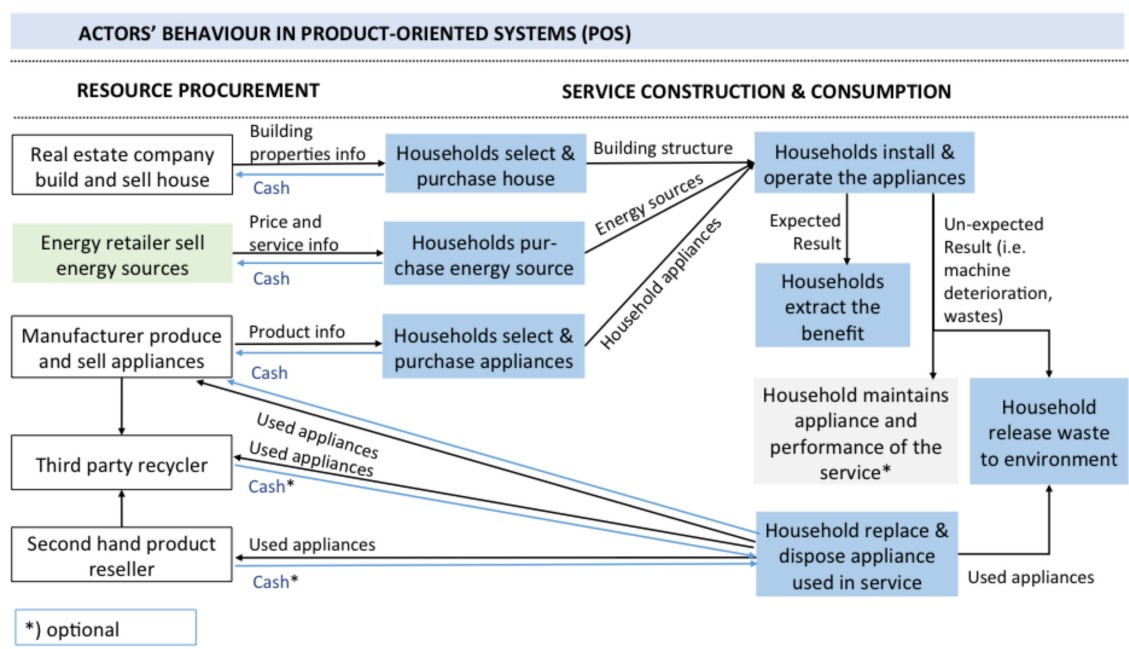

**Figure 2.** Actor's behaviour in POS.

In EPSS, the service provider focuses on providing the unit function derived from energy consumption, thus customer barely does anything except to receive the benefit of the service and pay for the unit consumption. Meanwhile the company conducts all tasks to transform the product and resource into something that can fulfil customers' need (see Figure 3).

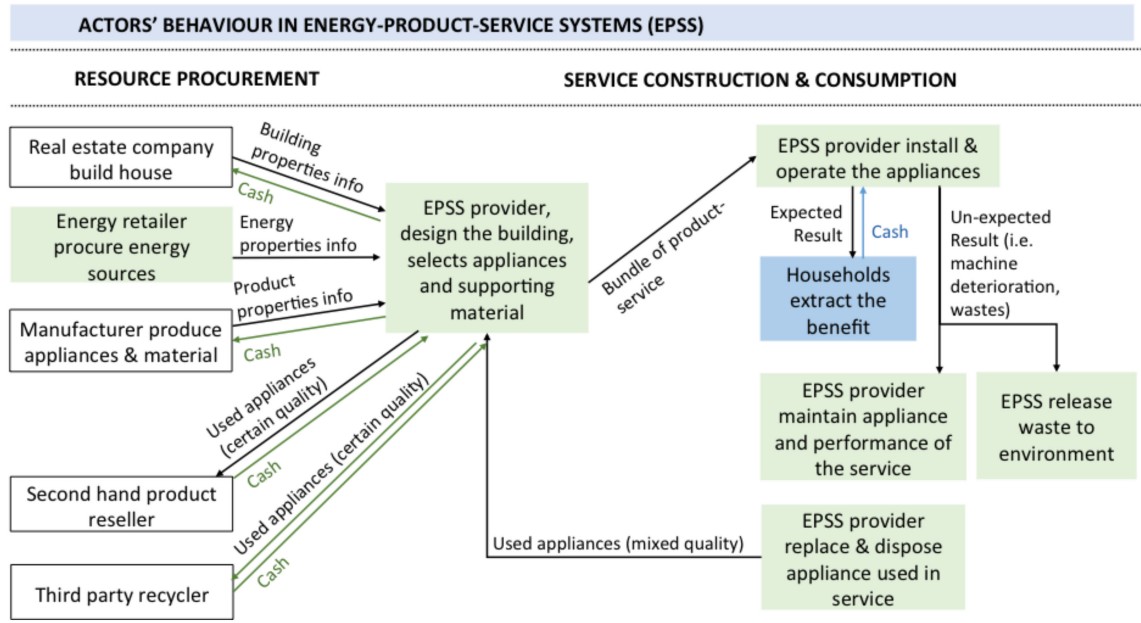

**Figure 3.** Actor's behaviour in EPSS.

From the perspective of design, with all due respect to its contribution to address environmental problem from industry, the proposed method in POS provides only part of the solution, due to its focus on products [5]. Each solution has its strength and also limitations to minimize environmental impact. However, when those methods are integrated into a system, sub-optimal solutions may occur and reduce the method's efficiency in addressing environmental issues [3]. Moreover, those methods emphasize on tackling upstream issues and pay less attention to downstream, while at the same time

industries are challenged with more sophisticated demands related to energy efficiency and material flows [6]. On the other hand, EPSS is designed considering problems in production and consumption, with the aim to improve resource productivity and efficiency in consumption through the combination of products and services to satisfy customers' needs [6–8].

## 2.2. The Difference between EPSS and Typical PSS

Energy is derived demand from economic activities such that it intertwines with broader social and infrastructural systems beyond just the energy generation and supply system. For this reason, we apply a system level approach to design business schemes targeted at households by considering customers' need and external partners as part of the energy system whose interactions are required to tackle related issues.

We determined that broader PSS exhibits similar ideas with EPSS in terms of the elements involved and purpose of the system. Various definitions of PSS [6,8–10] mention system elements included in the system, comprised of products, services and networks (of providers), which implies that the system involves multiple actors and components to create and deliver a combination of products and services. PSS was also introduced as a method to address environmental issues and economic sustainability simultaneously [11].

Despite the similarities, the major difference between EPSS and PSS lies in the system entities. PSS is envisioned primarily for manufacturing product [6,12–16] in which the resource input can be observed clearly. Meanwhile, EPSS considers energy as the core entity of the service as enabler for material or product operation. As the consequence, it should manage overall intangible of and tangible resources.

PSS focuses on increasing material productivity in production systems through alternative scenarios of product use and therefore, reducing material consumption. At the operational level, its ultimate goal is to close the material cycle by implementing product ownership shifts from customer to service provider (e.g., [5,6]). Vercalsteren and Geerken [17] provided extensive examples of PSS for households, which reveals that PSS tends to focus on a single product in designing the service, such as carpet leasing, renting toys and laundry services. Therefore, the network involves only the specific product/service provider and customer.

Regarding energy services, only a few studies have proposed PSS framework designs for energy services [18–20]. Energy was presented as a case study to show that service-oriented models can reduce the ownership costs of renewable energy power generating facilities, thus supporting energy accessibility in remote areas [18,19]. Another study designed PSS by offering ancillary services in addition to energy supply service to reduce emissions for industrial customers [20]. Despite proposing different ideas of PSS design, all of these studies emphasized on designing PSS for electricity supply service in systems separated from the broader system. Nonetheless, those frameworks exhibit typical PSS design in which the focus is on improving energy supply services without considering the broader energy system.

On the other hand, given the nature of energy system, EPSS have broader scope involving different suppliers. A simple energy system in a household, for example space heating system, requires at least an energy provider, heating machine provider, heating service provider and household as customer. A more comprehensive system might include a housing construction company, house leasing company and policy makers in the network of a space heating system. The extensive network of actors results in higher risk and most likely higher investment. However, the anticipated efficiency and environmental benefit is also greater.

## 2.3. The Difference between EPSS and Energy Services Contracting

There is another parallel discussion in the field of energy, commonly referred to as Energy Services Contracting provided by Energy Service Company (ESCO). These are alike in the way that the business integrates the energy consumption system to provide immediate results of energy

system performance to customers [21–24]. However, EPSS that is characterized with shifting resource ownership distinguishes it from the common conception of an ESCO. The shift of appliance ownership is a critical factor for EPSS because it is expected to boost customer economic benefit and enable the producer to better-manage maintenance, recovery and operation in order to improve profitability while increasing market competitiveness. This is not common for ESCO, which often have high project investment requirements that make it hard to gain wider market dissemination [22,24].

## 2.4. The Benefit and Barriers of PSS/EPSS Implementation

Although PSS is widely claimed to be environmentally and economically beneficial, the system is not by default have better performance compare to incumbent system [25,26]. It arguably produces a wide range of benefits in terms of material and cost efficiencies [6,25,27–29] as shown on Table 1. Nonetheless, stakeholders' readiness to adopt and implement is more likely to hamper PSS penetration in the market (shown on Table 2). Cultural shifting in consumer behaviour is suspected to be one of the barriers, as customers appear to be less enthusiastic about ownerless consumption [6,27]. Regarding companies, they are more concerned with their capability and the apparent difficulty of the organizational transition required to deliver combined product-services (e.g., [6,25,30]). Furthermore, the uncertainty of the benefits for the society hampers its adoption [6,30].

**Table 1.** Summary of PSS (Product-Service Systems) benefits.

| | PSS Benefits | References |
|---|---|---|
| Customer | Improvement in total value and quality; greater diversity of choices; personalized and customized offers; release from ownership responsibilities; lower cost and reduced problems associated with product ownership | [6,25,27,28,31] |
| Company | Creating competitive advantage; opportunities for innovation; increased market development; increased operating efficiencies; better feedback on consumer needs, locking-in customers, differentiation, revenue increase | [6,25,27,28,31] |
| Society | Reduced waste; reduced resource usage; closing material cycles | [6,27] |

**Table 2.** Summary of PSS barriers to implementation.

| PSS Barriers | References |
|---|---|
| Consumer related | |
| • Consumers not enthusiastic about ownerless consumption; lack of engagement and awareness related to PSS | [9,27,30,32] |
| Company related | |
| • Firms concerned about capability and infrastructure which are assumed to need high investment; lack of expertise in designing and delivering services; organizational changes | [6,25,27,30,32–34] |
| Society related | |
| • Socio-environmental benefits not always significant; uncertain profitability for company; unclear benefit for consumer | [6,30] |

To tackle those barriers, it is necessary to provide supporting evidence about PSS and EPSS performance. The potential of economic benefit may be a fundamental requirement for the stakeholders as the first step to introduce EPSS. Previous studies proposed methods to evaluate PSS performance

but it was mostly for environmental performance (e.g., [26,35–38]). There are only a few methods that considers to evaluate the economic benefit of PSS [39,40].

Meanwhile for EPSS, to find empirical evidence about its performance is challenging. There is no previous example of the service or experience to be observed or evaluated. Moreover, EPSS is different from PSS because it considers energy as derived demand and includes households' interest as the core of the service provision. Thus, there can be abundant alternative EPSS services constructed from combination of appliances and equipment as well as supporting service to deliver EPSS service for the household, whereas at the same time, the performance of those services is uncertain. For these reasons, this study deploys Simulation-Based Design to evaluate the economic performance of EPSS service.

## 3. Research Methodology

This study introduces a concept to address business competitiveness issues of current energy providers in a liberalized energy market, namely EPSS. A method to design EPSS service and evaluate the performance through Simulation-Based Design (SBD) is proposed. SBD is used to construct EPSS service considering stakeholders' interest and to evaluate the service performance, in which conducted through simulation of alternative EPSS service scenarios in order to seek conditions that are expected to fulfil stakeholders' requirements.

The problem with constructing EPSS service is there are no previous data, examples or experience regarding EPSS to be observed to formulate the service construction. Moreover, EPSS in general mimics social construction that involves human interests and their properties, which interact in creating complex behaviour. To identify which interest will lead to which behaviour and to predict which behaviour influences which performance can be very tricky. Interpreting the actors' behaviour and their interactions into a simulation model, as well as to streamline the system entities for the purpose of clarity requires tedious evaluation. The method is anticipated to overcome those challenges.

The present study develops SBD to design EPSS services and measure the economic benefit when competing with conventional business model in POS. First of all, a case study is presented to give a clear description of EPSS in the real world. The case study is also useful for defining the parameters and variables of the compared systems. In the present model, energy product/service under POS and EPSS is compared through space heating service system for households. The next step is to determine the goal of the service design as well as the performance criteria. And accordingly, simulation is developed to design EPSS and compare its performance with incumbent service under POS.

SBD consists of two main steps (as shown on Figure 4):

(1)    Service technical generation.

In EPSS, space heating service not only just providing space-heating system but also the service that accompany the resources usage. Thus, the first step of technical generation is to determine service configuration that construct the service. In EPSS, electricity retailer provides space-heating service, thus decision variable is to determine resource and service configuration that will minimize marginal cost from service generation. Given the alternative of service features the present method deploys brute-force approach, allowing spanning over all possible free combinations of service feature.

Resource requirement is determined according to service configuration, considering the resource specification, price and individual household's properties. Genetic Algorithm is used to find optimum configuration from all possible combinations of resources, specification and price to fit the constraints for each household.

(2)    Simulation-Based Design for performance evaluation and analysis.

This step is aimed to investigate the effect of parameters variation into system performance. Present study focuses on analysis of service and resource configuration variations effect on economic performance given set of market conditions and stakeholder's interaction in hypothetical liberalized

energy market. Evaluation of EPSS performance and POS is conducted using agent-based simulation when interact with variety of consumers' properties and behaviour in the market.

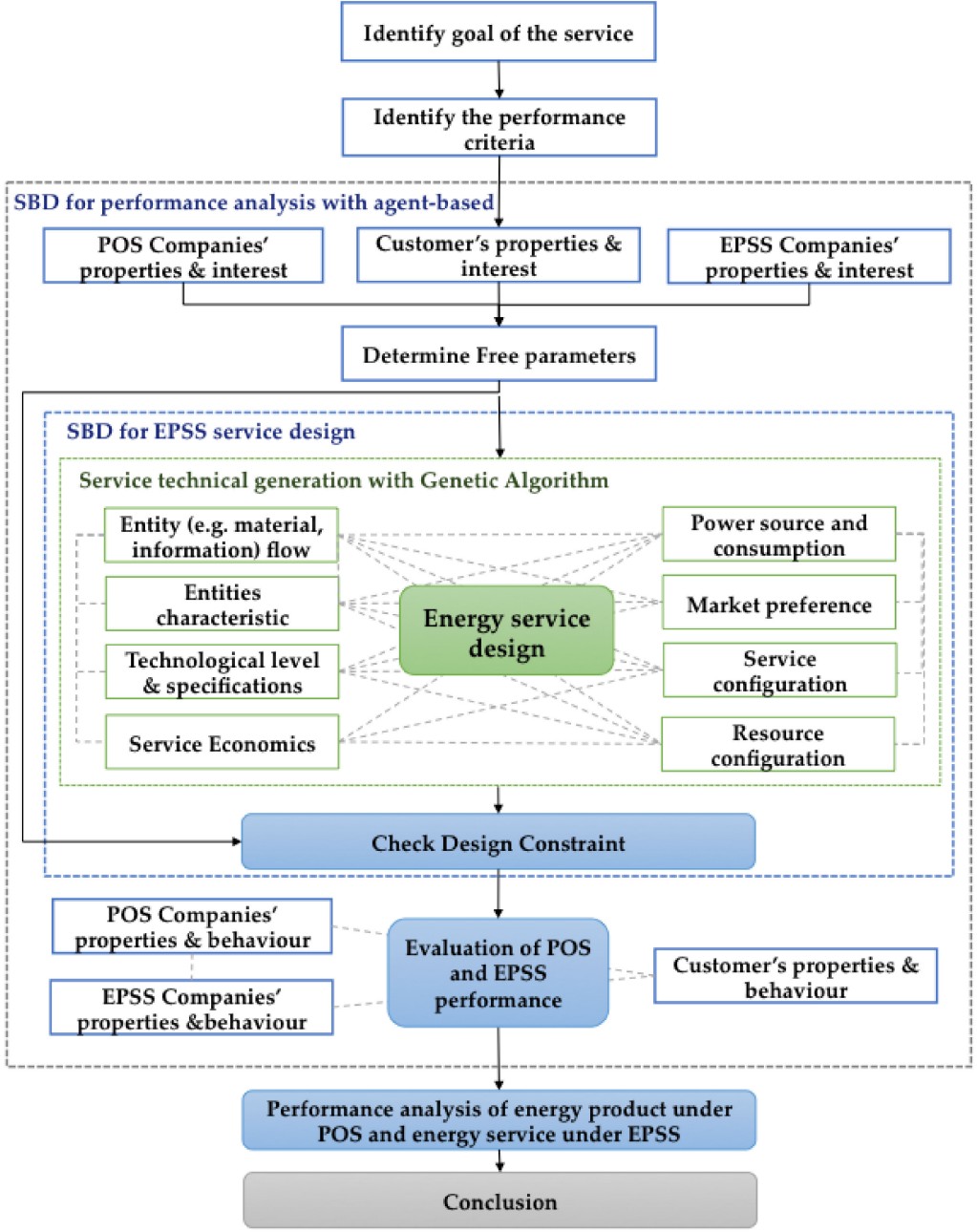

**Figure 4.** Simulation-Based Design

## 4. Case Study: Space-Heating Service for Households

Particularly in Japan, energy spent for residential space heating makes-up up to 10% of the total energy consumed as exhibited on Figure 5. Home heating energy use also contributes to 20% of the greenhouse emissions. Therefore, reducing energy consumption from home heating may result in significant monetary savings and emissions reduction. For this reason, this study selects home heating services as the case study under POS and EPSS schemes.

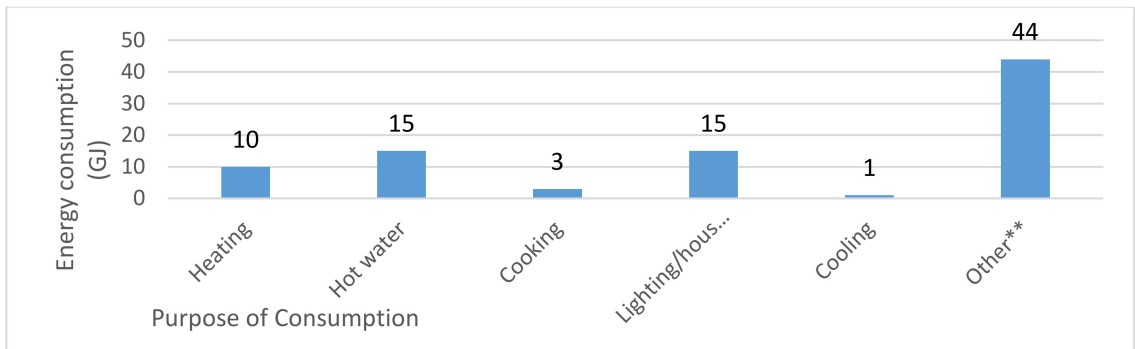

**Figure 5.** Japan Annual Energy Consumption per Household* (Year 2008) Source: [41]; * Households observed consist of two or more people; ** Other refers to purposes other than heating.

*4.1. Market Overview*

EPSS economic benefit is tested against virtual liberalized market. Numbers of consumers and producers freely enter the market and try to make a transaction. Consumers aim to maximize their utility considering their budget constraint, while producers maximize their benefit in subject to production constraints. Consumers who are willing to pay for the product and service according to the market price will enter the market, while some other do not.

Consider an electricity retailer wants to offer space-heating service for a group of households living in apartment block in Kansai area, Japan. Winter in Kansai is considered mild temperature since normally snows not lasting most of the time during winter. The minimum temperature can reach $-2\,°C$ and average temperature is $6\,°C$.

The households are living in buildings built from reinforce concrete structure with interior insulation (heating resistance coefficient is 1.1 in average [42]). The apartment block consists of fifty units, which were designed as compact apartment range from around $10\,m^2$ to $45\,m^2$. Due to the small size of the room, usually single individuals rent these units.

Initially, those households are connected to an electricity company to operate space-heating system. In these apartment units, heaters with electricity are mostly used on the designated apartments. Electric heat is delivered through heat pump units mounted on the walls or ceilings. Given EPSS offering, households are then given freedom to select whether they're going to remain with current system or switch to EPSS Company.

*4.2. Households' Properties and Behavior*

In space heating service consumption, decision-making process for consumer exists before and after purchasing the service. Decision before purchase occurs when household need to select and buy suitable heating device including energy source for heating service. After purchasing, one requires to make decision in consumption, involving the warmth level and time period to operate the system in respond to outdoor temperature. In current system, households make decision both before and after purchase [43]. However, in EPSS, households only make decision whether to purchase EPSS service or remain with current system. Consumers are not required to select the heating appliance or make decision regarding system operation, since EPSS Company takes over those processes.

4.2.1. Decision before Purchasing

The Internet of Things (IoT) has profound influenced on consumer behaviour. Individuals who are exposed to more information tend to becoming well informed, connected and empowered, creating a smart consumer who initiates to develop their definition of value toward product or service [44,45]. As the result, consumer value shifted from product centric to more personalized value. Not only that it alters consumer value, to some extent, it also helps consumer to be more rational in making purchase decision to optimize their benefit from consumption.

When households make purchase decision between POS or EPSS service, generally they want to maximize their utility by getting 24 h of comfortable indoor temperature every day. Yet, the effort of utility maximization is limited by cost constraint that they set for them self. Therefore, consumers can only optimize their benefit in subject to cost constraint.

Typical economic theory suggested that human decision making and behaviour are based on rational choice, presumably that consumers understand their preference and always aim to maximize benefit based on full and relevant information considering the cost constraint [43,46,47]. Accordingly, the present case assumes that consumers make purchase decision rationally. Moreover, consumers are categorized into two groups according to their preference, comprises of Conventional consumers and Environmental-oriented. Such preference eventually affects customers' willingness to pay for an EPSS. Conventional customers basically represent the group of customers who think that product ownership is important, while environmental-oriented customers represent the group of people that have environmental awareness and prioritize environmental performance in consumption.

Figure 6 exhibits customers' selection mechanism considering those conditions. Customers from all categories have a common demand for space heating. The ownership-oriented customer is bound to select the incumbent system to assure him/her getting the suitable design AC that matches with their taste. They select an AC and check the price to confirm that it is suitable to his/her willingness to pay. If the price is within their budget, the customer purchases the selected AC and at the same time, subscribes to the electricity supply service. However, in the case where the selected AC price is over budget, the EPSS service price is evaluated. But if the EPSS price is over their budget, the customer will eventually select an AC at an affordable price.

Customers from the environmental-oriented group also perform similar procedures in making their choice. Customers with this preference tend to select EPSS as they are more conservative on energy use [48]. Environmentally concerned customer initially selects EPSS and assesses the estimated EPSS rate. Note that customer willingness to pay for EPSS service from this group is higher than the other group. If the estimated EPSS service rate is higher than their willingness to pay, they will switch their selection into POS.

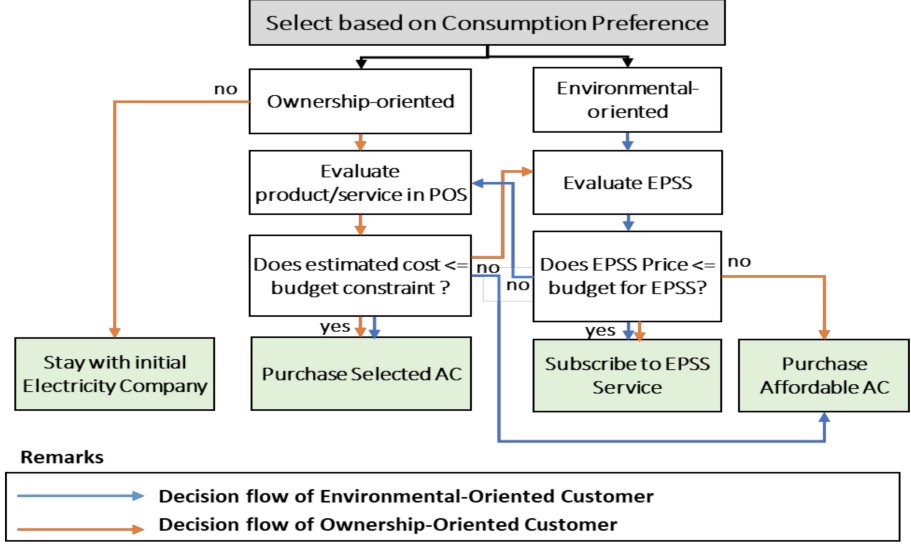

**Figure 6.** Customer Selection Mechanism.

### 4.2.2. Decision after Purchase

Not only households make decision before purchase, they also require making decision once consumption is started. Given electricity or service price per unit, households determine the quantity of consumption. Under POS, household increase their demand up to the level at which their marginal

benefit for electricity consumption is equal to the price they have to pay [49]. If the electricity price is flat, household consumption will only be affected by the cycle of their activities. Nonetheless, if the electricity price is fluctuated over the time, households mostly not interest to respond to it. Even if they were, the cost for infrastructure will reduce the most if not all the potential benefit for households [49].

Meanwhile in EPSS, company manages decision in consumption considering consumer's expectation about the service performance. Thus, it is more likely that company will take the most efficient measure to minimize cost.

## 4.3. Company's Properties and Behavior

POS and EPSS use different approach and behave differently in respond to households' demand toward comfortable room temperature. In this study, both POS and EPSS Company invest on renewable energy generation and aim to get the benefit from it. Both invest on electricity generation from solar power. Electricity retailer under POS offers 'green electricity' to supply households. Meanwhile, EPSS Company offers space-heating service unit for the household. As consequence, EPSS company needs to invest on appliances and resources accompanying electricity to deliver the service. Before proceeding to detail information about space-heating service under POS and EPSS, first we provide a description of space-heating service in general.

### 4.3.1. Space-Heating Service System with Air Conditioner

Studies have been done to achieve highly efficient heating system for households to achieve cost efficiency and reduce emission level [50]. Ideally space-heating system is designed through several interrelated process, comprises of draughts-proofing and insulating, selecting or improving heat distribution, selecting heating energy source and selecting heating equipment [51].

The advantages of draughts-proofing and insulating is to achieve efficient cost from heating consumption, so that indoor temperature tend to be warmer on winter. Fewer draughts and good insulation also maintain expected humidity levels because dry air on winter is caused by too much air from outside getting in the house. It is safe to say that to increase humidity level and at the same time, lower heating cost is by reducing air leakage [51]. However, installing mechanical ventilation system without causing draughts is also necessary, to keep the air quality inside the house. Accordingly, additional cost is needed to get efficient space heating system.

In term of selecting heat distribution system, it depends on various factors, such as cultural aspect, size of the space being warmed up and economic consideration. In some countries the heating system includes distribution system, while other countries (such as Japan) households prefer to use space heater that does not need distribution network. Thus, as previously mentioned above, we focus on the use Air Conditioner with heating system. This appliance uses electricity as energy source. It works by providing heat directly to the room in which they are located, thus does not require distribution system.

However, households are more likely to perceive that comprehensive consideration in space heating design is costly because they must hire a consultant for efficient design and put more investment into resources and material to get a highly efficient heating system. Avoiding the hurdle, households choose a simple way by ignoring the efficient design and conserving energy by limiting heating by keep it on lower temperature or shortening the time use. To handle such issues, heating equipment producer add the efficiency of the equipment for the households that can afford a highly efficient machine as a way to conserve energy and achieve cost efficiency. Therefore, there's no guarantee for any company that offering a comprehensive design of heating system will get benefit from the market. Accordingly, this study evaluates the cost-benefit performance of space-heating service for service provider under EPSS framework in liberalized market. The analysis considering free parameters the service design which is constructed from two sets of technical aspects: (1) Resource (product) specification and configuration, (2) Service feature and configuration.

### 4.3.2. Resources Specification and Configuration

It was determined that key appliance of the space-heating system is Air Conditioner, both for service under POS and EPSS. In POS, households select and purchase AC in the market that available in various specifications, differentiated by efficiency, coverage area and price. We assumed that given the same coverage area, the higher the efficiency, the more expensive the price. Electricity demand in POS depends on the initial building heat resistance and appliance efficiency, given that households heating system only consisting of AC.

Meanwhile, in EPSS, beside Air Conditioner, electricity retailer completes the service with supporting appliance and resources comprises of additional insulation material, demand response programs and wireless sensors network which connected to GPS on smart phone and PV Panel. Given the alternative of resources (shown on Table 3), service provider determines the configuration with the aim to optimize benefit and minimize cost.

**Table 3.** List of appliances to deliver space-heating service.

| Appliances/Material | Remarks |
| --- | --- |
| Space heater (Air Conditioner) | Differentiated by its coverage area, efficiency and price |
| Draughts-proof and insulation material | Differentiated by wall area coverage, heat resistance coefficient and price |
| Wireless sensor network connected to phone GPS and Demand Response Programs | Only for EPSS operation control |
| Renewable Energy Generator (Solar PV) | Only for EPSS operation control |

### 4.3.3. Service Feature and Configuration

The service in general covers appliance selection and procurement, heating system operation and maintenance. Various tasks should be conducted to manage the service across its life cycle as shown on Figures 7 and 8.

*(a) Service in POS*

In POS households pays for the product (shown on Figure 7) and perform all service tasks by themselves to get the expected benefit from the product.

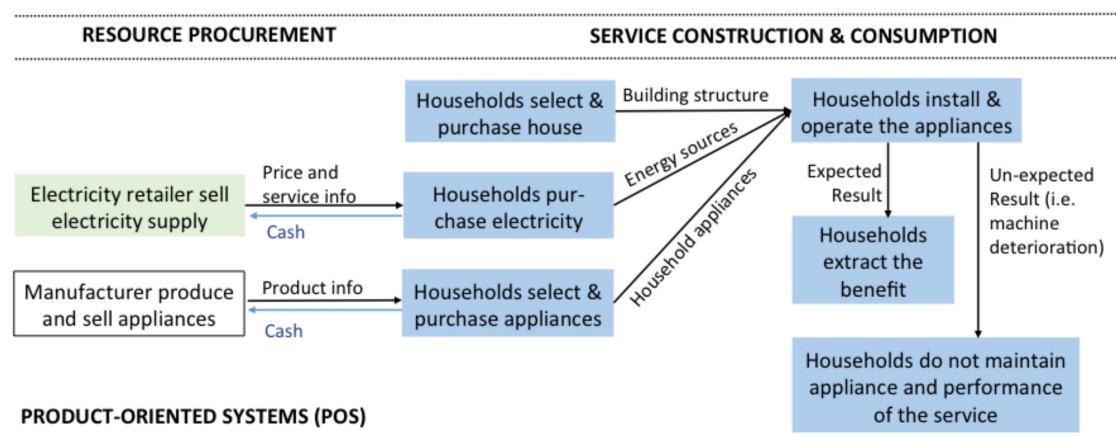

**Figure 7.** Product/Service and Cash Flow in POS.

The role of electricity retailer in POS is to supply electricity for households. Despite that households are hardly respond to price changing, electricity company applies dynamic electricity price in attempt to balance electricity supply-demand. Electricity price during peak demand is higher than average demand.

*(a.1) Appliance Selection*

In POS, households select the appliance by itself. Typical POS households in the system only select and install Air Conditioner to warm the room and that insulation material follows the standard heat resistance required with Japanese government [42]. Air Conditioner is chosen considering price, personal taste (such as appliance colour, design and technical specs) and by approximating AC coverage area.

This approach gives households freedom to choose the appliance based on their personal taste. However, lack of technical knowledge of space heating system may cause missed design, in which the selected appliance not suitable with the technical requirement for specified room or building. Unfortunately, households cannot return and change the specification of appliance after installation. Households have to remain with their selection until end-of product life cycle.

*(a.2) Appliances Operation*

Households that purchase electricity supply under POS operate the appliance by themselves. One turns on Air Conditioner when the room temperatures felt too cold according to his/her feeling. Then he/she has to wait for a moment before getting the comfortable temperature as needed. When it is not needed, the air conditioner is turned off manually.

*(a.3) Appliances Maintenance*

Households have tendency to not perform regular maintenance. They perform maintenance when operation abnormality is detected or when the machine is broken. However, most of the time, abnormality is hardly detected. This study considers that households only perform maintenance when the machine is broken. Thus, lack of maintenance leads to earlier deterioration and inefficient machine performance.

*(b) Service in EPSS*

Households in EPSS pays for the result of energy service unit, therefore service provider performs all the tasks from selecting appliance and performs maintenance for appliance and service performance. Hence, several alternatives of EPSS services considering the sub systems of the energy supply demand system are introduced for space-heating service under EPSS. From the basic characteristics (i.e., consumer ownerless tangible resource), it is extended into several service designs that are distinguished by combinations of its feature, consists of: (1) *Operational System Design*, (2) *Consumption Management and* (3) *Electricity Supply Management*. As the result, energy service provided by EPSS appears to be more expensive in compare to POS but with less task and responsibility for the customer, as shown on Figure 8.

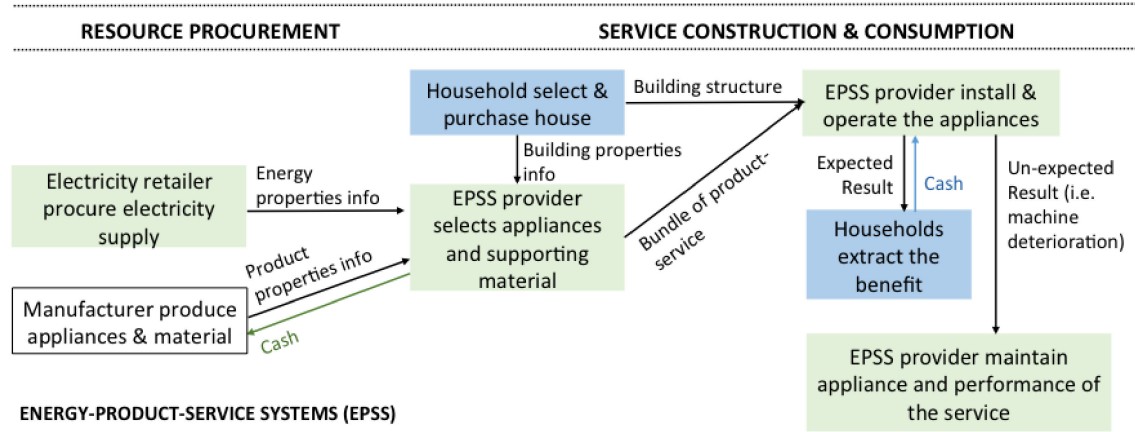

**Figure 8.** Product/Service and Cash Flow in EPSS.

*(b.1) Operational System Design*

EPSS Company determines the resource configuration to deliver EPSS service. Given the household's state (i.e. the wall area, construction and material of the building, space heating demand occurrence in term of temperature and period and household's budget for the service), EPSS determine resource configuration for the households. Therefore, the system operation is expected to be more efficient in term of electricity consumption. However, households do not own the appliances and they cannot select appliance that suits to their taste.

This feature also aims to maximize energy efficiency by designing the physical environment of energy consumption. Managing the physical system of electricity consumption is conducted by selecting the optimum combination of Air Conditioner efficiency and dwelling heat resistance. Customer's willingness to pay and the service price are taken into consideration in deciding the optimum choice.

*(b.2) EPSS Operation*

In the system operation, EPSS Company introduces Service Consumption Management and Electricity Supply Management.

Regarding *Service Consumption Management*, the purpose of this feature is to manage electricity consumption through a Demand Response Program. Given the available electricity supply in the system, if the electricity consumption rate is higher than a determined threshold, electricity price is set higher to control the demand. Accordingly, company will adjust electricity consumption by reducing heating temperature to a suitable temperature range as recommended by WHO [52] to manage electricity cost at efficient level.

The company controls the space-heating system using wireless sensors network to know when and where to turn on and off the system, as well as to set the temperature level. The system is programmed so that GPS signalling the system to turn on the AC when the resident is located near the house, thus the room comfortable room temperature is ready when he/she arrives home. The system setting turns off the AC when nobody is home.

*Electricity Supply Management* manages electricity supply by installing PV Panels to supply electricity for the heating service. In this model customers do not bear any cost associated with PV Panel usage and installation. However, the company is given permission to utilize dedicated space, free of charge, on or around the customer's property for PV installation. The electricity generated from the facility is then used to supply the household's heating demand.

In this model, electricity consumption is included as an operational cost for the service. Customers who use this service feature benefit from lower electricity prices than the market, per kWh. Hence, it is expected to lower the EPSS service price.

In the case of electricity shortage from the PV panel, the company supplies electricity from the grid. However, if there is an excess of electricity, the company has the right to sell the electricity to the grid, which is automatically included in the company's profit.

*(b.3) Appliance and System Maintenance*

EPSS system is completed with real time performance checking to detect if there's abnormality on the operation system, such as leakage and noise. EPSS Company conducts regular maintenance to control appliance performance deterioration, which is expected to extend the appliance's effective life span.

Eventually, the energy sub systems' design and control can be proposed as a service feature that may be added or removed from an EPSS service package. From customer's perspective, there are eight alternatives of service designs available, while from the view of companies, there are three scenarios to create revenue and improve their profitability. EPSS designs constructed from combinations of the features are exhibited in Table 4.

**Table 4.** Simulated EPSS Service Designs.

| EPSS Design | Maintenance | Managing Electricity Consumption | Operational System Design | Managing Electricity Supply |
|---|---|---|---|---|
| 1 | V | - | - | - |
| 2 | V | V | - | - |
| 3 | V | - | V | - |
| 4 | V | V | V | - |
| 5 | V | - | - | V |
| 6 | V | V | - | V |
| 7 | V | - | V | V |
| 8 | V | V | V | V |

## 5. Simulation Development

### 5.1. Model Description and Assumptions

Simulation model is developed to assess the economic performance of EPSS approach compared to POS in a liberalized electricity market. It simulates electricity and service generation to satisfy household's space heating demand in hourly manner. Economic benefits include customer and company's benefit, as well as customer satisfaction is evaluated during 2 years' market activities for each system design including POS.

The model is developed with the assumption that half of the customers in the market are environmental-oriented and the rest are ownership-oriented customers. Customer's willingness to pay for energy product and service is constant over the time, with ownership-oriented customer has willingness to pay equal toward POS product and EPSS service. On the other hand, environmental oriented customer has willingness to pay for EPSS service 20% higher than POS service. The household appliance used in POS and EPSS both are on the same level of technology. However, due to bulk purchase in EPSS, the company get discounted price from the manufacturer, thus the appliance cost is 30% higher for households.

Both POS and EPSS supply electricity from the main grid applying same electricity rate. Electricity rate is distinguished into two level depend on the load. On the event of peak electricity demand, the price is set higher to control the demand. The rate is normal when electricity load is below peak demand threshold. We define peak demand as situation when electricity demand at the time reaches 80% of electricity supply. Under the PV scenario, where EPSS includes self-generated electricity as part of the service, the electricity rate generated from PV is considered free, yet service and maintenance costs are included as service expenses. All PV generators and loads are connected through lossless network of infinite capacity. Thus, generated electricity from solar power is 100% delivered for households' consumption without any loss. For PV Panel, there are hundreds of types PV panel provided by manufacturers. However, we chose one specification for this model for simplification. Solar panel is made from Monocrystalline, with efficiency 16% and performance rating 75%, with cell size 125 × 125 mm. Installation and operation cost of PV panel are based on estimation by METI as much as 325,000 yen/kW and 10,000 yen/kW consecutively [53].

The model also assumes that during two years' evaluation all parameters is fixed and are not affected by any changes on external stimulus. Space heating demand occurs stochastically in term of temperatures and occurrence depends on the weather and customer's lifestyles. Given the demand, customers then make the decision to select between POS product and EPSS service. After engaging with one system, customers consume the product or service and pay for it. Customers evaluate the benefit and satisfaction during the evaluation period without a chance to switch to the other system. At the same time, companies also evaluate the economic performance without any chance to change into another system.

*5.2. Identification of Performance Criteria*

In the case of space-heating service, the goal of the service design is to provide space-heating considering stakeholder's interest. Accordingly, these interests are taken into consideration for identification of performance criteria. Naturally the actors want to ensure that their participation in the market will provide them economic advantages over other benefits. For this reason, this study specifies the economic benefit for customers and companies as a key criterion for comparison.

Customer benefit is evaluated based on the amount a consumer is willing to pay for the product or service minus the price that the consumer actually pays. This measures the benefit that customers receive from participating in the market. Customer satisfaction is also evaluated as the ultimate goal of customer's benefit. Customer satisfaction is evaluated considering individual preference toward electricity product/service. Thus, economic-oriented customers and environmental-oriented customers may share or may have different expectation toward product/service performance. In this model, both types of customers expect to have an economic benefit and to maximize space-heating utility according to their needs. However, environmental-oriented customers expect to have lesser environmental impact from their consumption, thus environmental consideration is included on their satisfaction criteria.

Meanwhile companies aim for profitability, in which calculated based on the product or service price minus production costs that company spends to produce and deliver the product/service. Finally, customer satisfaction is predicted by the level of perceived performance conforming customers' expectation. The detailed explanation of each criterion is clarified below.

5.2.1. Economic Benefit

The calculation of economic benefit in this study follows the definition from Mankiw [47] of customer's and producer's benefit from engaging in a market transaction. Customer benefit for each customer—$i$ is evaluated based on the amount a consumer is willing to pay ($W$) for the product or service—$j$ minus the price ($P$) that the consumer actually pays. It measures the total economic benefit Y in ¥ during evaluation period that customers receive from participating in a market, which is formulated with

$$Y_{ij} = W_{ij} - P_{ij} \qquad (1)$$

In this study, the EPSS Company approximates service price relative to the customer willingness to pay. Accordingly, the price effect on the customer's decision to select between purchasing POS or EPSS is evaluated and analysed.

The customer's economic benefit strongly depends on the customer's preferences, which influence customer willingness to pay for POS and EPSS. Therefore, the result is very subjective across individuals. A customer's actual expenses for product/service consumption is estimated to objectively compare the customer's economic benefit between POS and EPSS.

To measure the benefit that companies gain from participating in the market, companies' profit is calculated based on the product or service price minus production cost that the company spends to produce and deliver the product/service. The company's profit is evaluated in short term, by comparing all EPSS scenarios and POS. In this study POS as the incumbent system has reached economic scale. Thus, POS profit is simply calculated by the margin for each unit product multiplied by the number of sales.

On the other hand, the EPSS price and production cost are estimated based on the POS figures. Assuming that there are numbers of customers that are willing to purchase the product/service, the customer decision to select between POS and EPSS is also expected to distinguish the company's profit. Hence we get the generic formula for Company's profit of product or service *j* as follows:

$$\Phi_{ij} = \sum_{i=1}^{N}\left(P_{ij-}\Gamma_{ij}\right) \tag{2}$$

where $\Phi_{ij}$ denotes company's profit and $\Gamma_{ij}$ denotes production cost of electricity product or service for each customer in ¥.

### 5.2.2. Customer Satisfaction

Customer satisfaction is the ultimate goal of marketing activity that is used to influence customer purchasing behaviour, hence it necessary to discuss the association of this factor with profit generation [54,55]. This study evaluates customer satisfaction after EPSS engagement to predict customer future purchase behaviour of EPSS.

Customer satisfaction has a long history of research. Various methods and frameworks have been established to predict customer satisfaction, either for specific cases (e.g., [56]) or general cases of services or products (e.g., [57]). Some methods simply use a happiness scale that assesses perception against expectation. The sophisticated ones elaborate customer reviews using big data analysis to predict customer satisfaction [56]. Nonetheless, major satisfaction assessments are built upon a disconfirmation paradigm assuming that satisfaction is related to one's initial expectation and the actual product/service performance [54,57,58].

In an attempt to introduce EPSS and compete with POS, customer satisfaction for both schemes is evaluated. Although both systems are essentially different, whereas POS provides products while EPSS delivers service, it is possible to compare the customer satisfaction [57]. Customer satisfaction is measured knowing that customer behaviour and perceptions of advantage are the antecedents of satisfaction and that expectations and performance have a direct and significant effect on one's satisfaction [57]. This study constructs customer satisfaction by evaluating three aspects encompassing economic benefit, service performance and household's preference.

Economic benefit is measured using Formula 1 in monthly period. Customer will economically satisfy if their actual expense is equal to or even lower than their budget constraint. However, pay lesser not necessarily completes customer happiness if the service performance does not match to their expectation. Regarding service performance, both customers group expect to get the warmth level according to their comfort level when they need it. POS customers have more freedom to set the temperature level at any time they need, because they have full control over space-heating operation system and for being insensitive toward price changes. On the other hand, EPSS Company takes over the operation system to optimize the consumption considering household's demand of warmth level and cost constraint. Through simulation, each household's demand, which represent his/her expectation, is confirmed with the actual service in which delivered at the requested time. It will satisfy customers if it is confirmed between expectation and the actual performance, not less or more.

While economic-oriented customers only focus on economic benefits and service performance as an indicator of satisfaction, environmentally-oriented customers include environmental consideration that can improve their satisfaction in consumption. Environmental performance is measured considering emission released from space heating consumption and efficiency of electricity consumption. Emission level and electricity consumption are compared to average households' emission and electricity consumption in the given market. In this case, IoT support the company to collect and perform calculation toward electricity efficiency and emission released from the market and accordingly provide the record of household's environmental performance from EPSS service consumption.

Electricity consumption for each customer—$i$ when consuming product/service—$j$ is the function of hourly heating required ($\mathcal{H}$) in *KwH*, appliance coefficient of performance *(COP)*, rate of system performance deterioration (σ) and operational time (*t*) on the given month. System performance

deterioration rate is influenced by system maintenance and appliance usage pattern. Accordingly, the equation of monthly electricity consumption (in *KwH*) is given by

$$E_{ij} = \frac{\mathcal{H}_{ij}}{(COP_{ij} * \varsigma)} * t \qquad (3)$$

Supposed that space-heating emission factor is 0.52 ($CO_2$.kg/KWh) [59] and given electricity consumption, hence we get the equation for monthly emission level for customer—*i* when consuming product/service—*j*

$$\exists_{ij} = 0.52*_{ij} \qquad (4)$$

Accordingly, customer—*i* satisfaction toward aspect *k* of electricity product or service *j* is predicted by the difference between customers' expectations and perceived performance $\wp$, hence satisfaction of customer—*i* is presented with

$$\mho_{ijk} = f(_k, \wp_k) \qquad (5)$$

Customer satisfaction value is presented as a binary value (0,1) implying confirmation or disconfirmation between expectation and perceived performance. The satisfaction value is 0 if there is disconfirmation between customer expectation and perceived performance, indicating that the customer is dissatisfied about the performance of the product or service. On the other hand, the satisfaction value is 1 if the customer is satisfied with the performance of the service, considering his/her expectation.

Eventually all the satisfaction elements are aggregated to estimate an overall satisfaction index, given by

$$\overline{\mho_{ij}} = \frac{\sum_{k=1}^{n} \mho_{ij}}{n} \qquad (6)$$

The aggregate value ranges from 0 to 1, where 0 value implies that the customer feel less satisfied, values between 0 to 1 imply fairly satisfied and 1 implies customers are very satisfied with the offers.

### 5.3. Simulation Objectives

Through simulation, this study attempts to provide supporting evidence regarding the common premises that are prevalent in PSS discussions as below.

**Hypothesis 1.** *EPSS design improves customer economic benefit*

This hypothesis is made based on the premise that PSS customers are released from the cost and risk of ownership such as purchasing cost, appliance maintenance and part or component replacement. In the case of EPSS, customers may get additional economic benefit due to higher efficiency in electricity consumption. Thus, the simulation is conducted to prove whether households will gain higher economic benefit through EPSS or otherwise.

**Hypothesis 2.** *EPSS design results in the same satisfaction level compare to POS*

This hypothesis is based on the premise that PSS results in improvement of total value and quality of service with greater choice. In addition, the quality of energy service is clearer and easier to be evaluated, thus it is expected to improve customer satisfaction.

**Hypothesis 3.** *EPSS generates more profit compare to POS*

With greater choice and better quality of service, EPSS offers higher level of service for the customer. The competitiveness is anticipated to be significantly improved together with appliance ownership shifting, releasing customers from the cost and risk associated to ownership. As a result,

market share is increased that leads to higher revenue and profit. It becomes the underlying premises for Hypothesis 3.

Design-based simulation has been established demonstrating POS and EPSS competition in the market. Market outcomes and system performance are extracted from the simulation as evidence for the hypothesis and to be analysed. We also evaluate the market outcomes that relates with the system performance and the hypotheses.

### 5.4. Simulation-Based Design for EPSS Service Design and Evaluation

Company's behaviour between POS and EPSS are different, such that Simulation-Based Design implementation for both systems is also different in term of input parameters and variables. In POS, resource and service configuration is considered as fixed parameters (see Figure 9). Resource used in POS is only Air Conditioner with certain level of price and efficiency. Furthermore, service cost in POS in zero, since households generate their own service. Hence, service configuration is ignored. In EPSS, resource and service configurations serve as free parameter, in which under different market conditions, will determine the cost and benefit for EPSS provider.

Simulation model depicting consumer and company's interaction is build using *Python Programming Language* and incorporate Genetic Algorithm to optimized operation system design in EPSS.

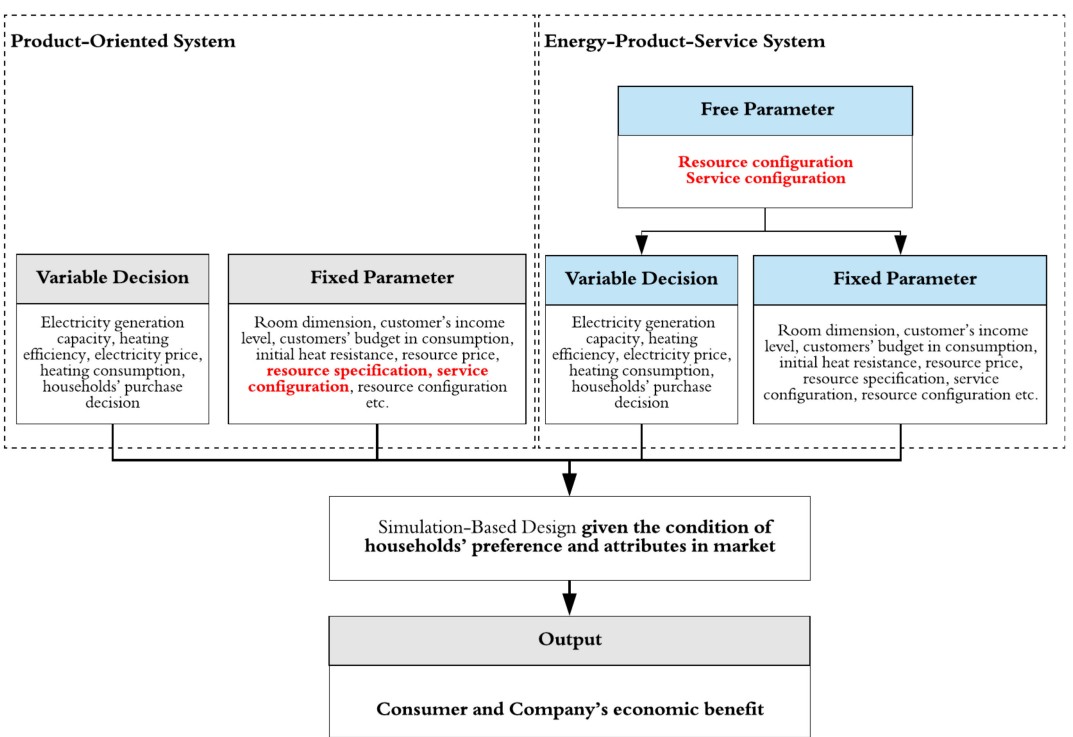

**Figure 9.** Simulation-Based Design to design and evaluate EPSS service performance relative to POS service performance.

### 5.5. Data Collection

We use the latest data available online to represent and approximate households and POS Companies' figures. Data associated with dwellings, the dwellers behaviour and weather are collected from secondary resources and interpolated from publicly available data [41,42,48,53,60–62]. For companies under POS, we conducted an online survey to collect information about available AC in the market, including specifications, efficiency performance and price. Regarding electricity supply service, all figures in this model are based on the latest figure publicly available from Tokyo Electric

Power Company (TEPCO). Eventually, data of EPSS consumers and company are approximated POS figures.

## 6. Simulation Results

The model described in the previous section is used to simulate POS and EPSS competition in a liberalized market and to measure the system performance. We capture the short-term pattern of EPSS performance, particularly economic performance and customer satisfaction. Under each customer's rational decision making to select between POS and EPSS, the result of the simulations is discussed as follow.

The first scenario and second service design emphasize the shifting ownership with additional demand response program for the second scenario. In these scenarios, customers are released from costs and problems associated with resource ownership. However, additional charges are set for customers in the form of service costs, which are approximated from appliance prices, maintenance costs and operational costs (electricity consumption cost).

The third and fourth service design includes operational system design feature and is advanced by including demand response program in the fourth scenario. Customers get advantage from highly efficient electricity consumption, thus reducing operational costs of the service. Service cost structure is set similar to previous scenarios, with additional investment for the insulation system.

The last four scenarios, including the fifth, sixth, seventh and eight service design are basically replication from previous scenarios but with additional PV panel installation to supply the electricity for space heating demand. Under these scenarios, electricity rate to power space heating demand is 30% lower electricity supplied by POS.

As the result, in average of 55% of customers select EPSS service for all EPSS designs, as shown on Figure 10. As much as 3% of that share consist of ownership-oriented customers, means that some customers select the service not in accordance to their initial preference. Customers switch their selection because the service price is more affordable than the appliance price that customers initially wanted to purchase. It indicates that the price set by EPSS Company in this simulation is competitive enough compared to the product price of the incumbent system.

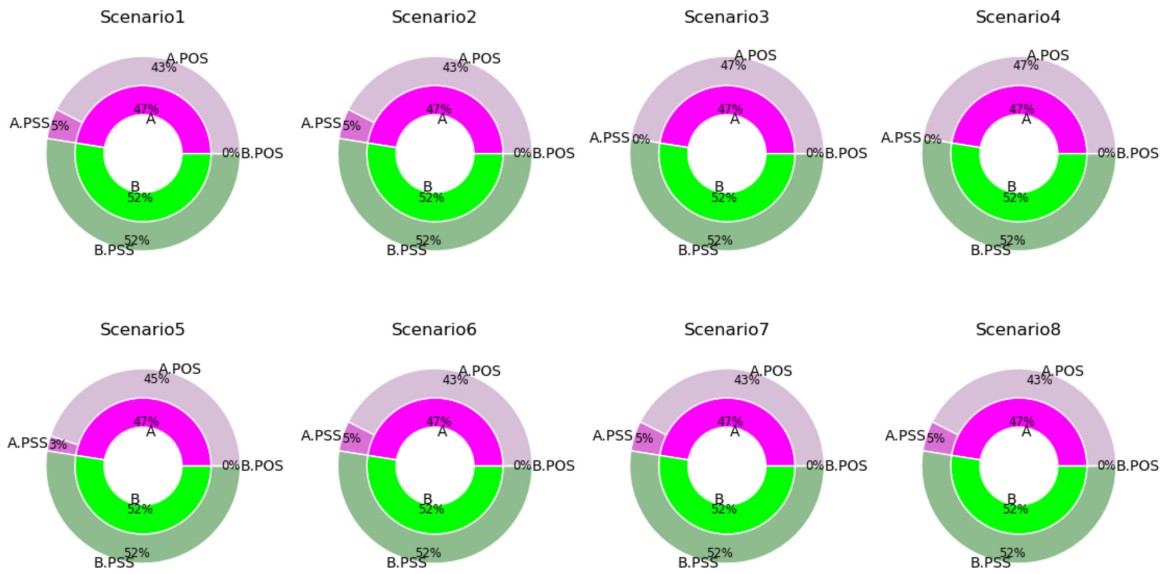

**Figure 10.** Market Share for All Scenarios6.1. Customer's Economic Benefit.

### 6.1. Customer's Economic Benefit

The model also estimates customer's economic benefit of EPSS scenarios in comparison with POS considering the willingness to pay and product/service price. The economic benefit is evaluated to

provide evidence for Hypothesis 1 stating that EPSS improves customer economic benefit because it releases customer from ownership cost. To complete to analysis, we also calculate the expense that customer spent for consuming EPSS service or POS product. Figure 11 exhibits the results after simulating the model for 2 years of EPSS introductions in the market.

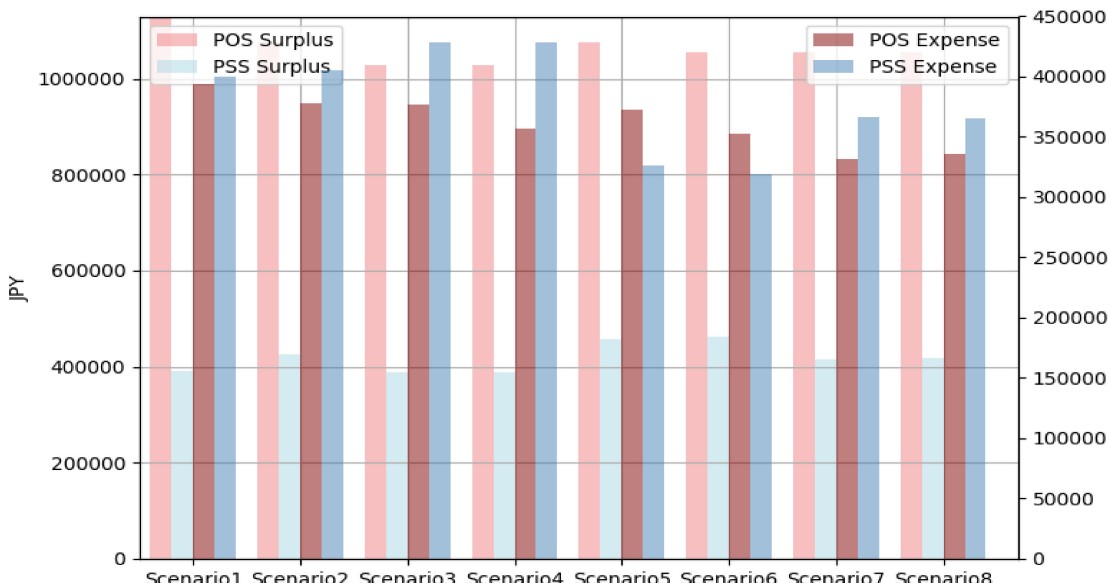

**Figure 11.** POS & EPSS Economic Benefit Comparison.

Before discussing the main topic of this section, please note that POS customer's economic benefit and expenses vary between all scenarios. Scenario 5 even shows that POS expense is slightly higher than EPSS customer's expense. Despite the model being set using the same parameters and conditions for all scenarios, especially for POS customers, the decision to select AC is randomly assigned. Such decisions influence the individual expense for AC purchasing which eventually determines the total and average expense for all POS customers in the system.

The figure shows the average of EPSS and POS customer's economic benefit and the actual expense spent for the consumption. Given the customer's willingness to pay relative to product/service price, customer benefit under POS is significantly higher than EPSS for all scenarios. It implies that in this model, customers that choose the POS product set their budget very high compare to the actual price. In reverse, EPSS service price is close to customer's budget for service expenditure. In this study, customer benefit strongly relies on the willingness to pay. Hence, there's no surprise in the results, which exhibit that customer's economic benefit strongly depends on the market preference.

Regarding the average expense, Figure 11 reveals that EPSS expense tends to be higher than POS expenses. However, there is possibility that it can be slightly lower than POS expenses as shown on Scenario 5. In addition, we first noted that the demand response program in this simulation does not significantly impact on expense reduction for all scenarios due to relatively low electricity demand considered in the system model, which is only from space heating demand.

The highest expense resulted from EPSS third and fourth design among all service designs. The service rate is higher due to additional investment for insulation system. Although electricity consumption is significantly reduced from operational system design, which results in lower electricity expense (shown in Figure 12), yet the investment to build the system is significantly higher compare to the efficiency, hence the service remains high.

The most cost efficient are the fifth and sixth scenarios because they provide lower electricity prices compared to the first four service designs, thus operational costs can be significantly reduced. Different to the seventh and eight scenarios that also include PV panel, their total expenditure for each customer is relatively higher. It is because these scenarios involve optimization of the physical system.

Electricity expenses are significantly reduced compare to all scenarios but are still more expensive compare to the fourth and fifth service designs.

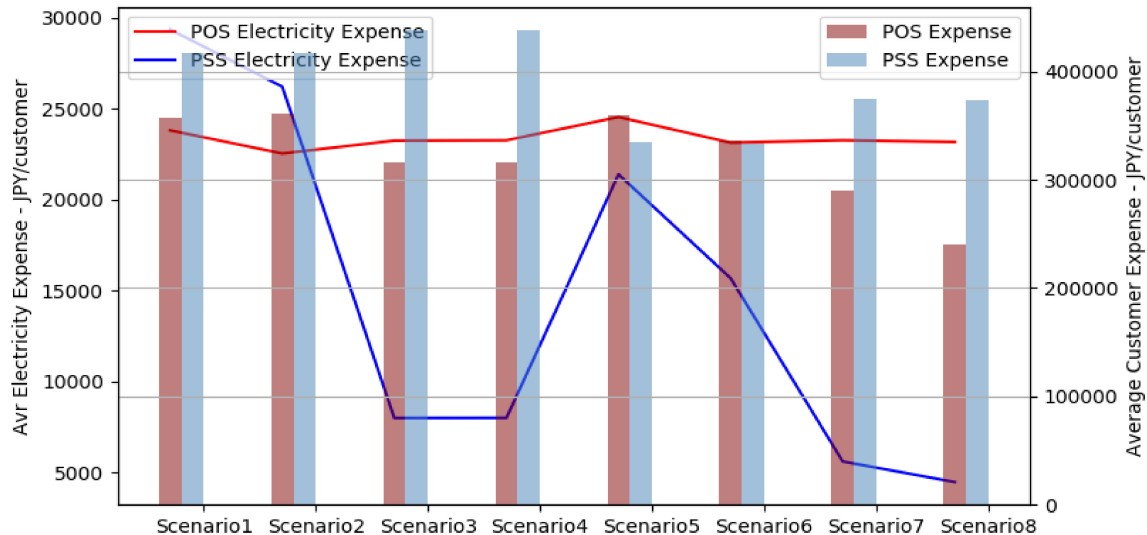

**Figure 12.** POS & EPSS Electricity Expense Comparison.

Returning to hypothesis 1, which was based on the premise that EPSS can improve economic benefit, the result of the simulation provides evidence that this hypothesis is rejected. Given the predetermined cost, price and customer willingness to pay, for short term contracts, the proposed EPSS designs hardly improve economic benefit for the customers and the total expense during contracts tend to be higher in comparison with POS. This may change with longer contracts or alternative considerations.

*6.2. Customer Satisfaction*

Customer satisfaction is tested to prove Hypothesis 2, whether EPSS service design may result in at least the same level of POS customers' satisfaction. Figure 13 illustrates the comparison of customer satisfaction between EPSS and POS under scenario 5. Three satisfaction factors that construct satisfaction in this model, encompassing economic benefit, preference and service performance are also included in the analysis.

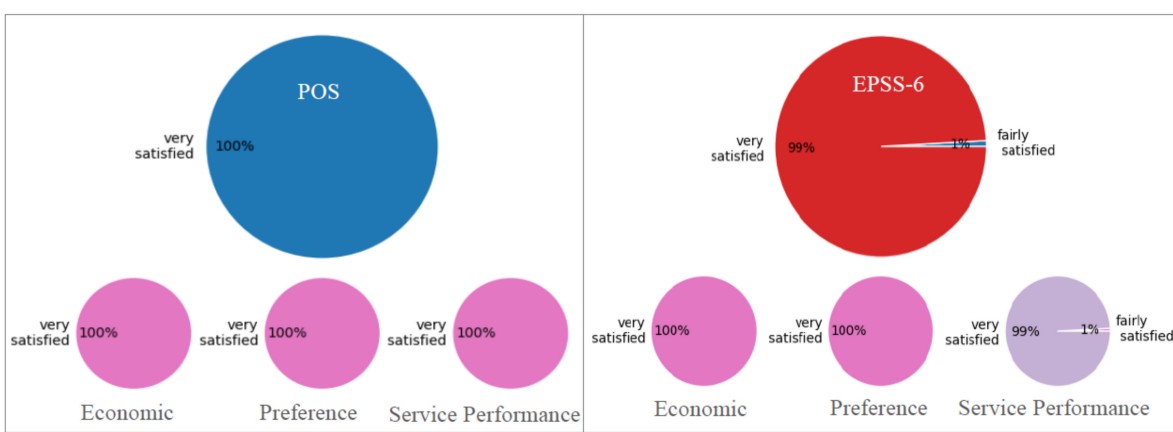

**Figure 13.** POS & EPSS Customers' Satisfaction Comparison.

The simulation results show that there is no significant difference between POS and EPSS across all scenarios. It is because some satisfaction in POS needs to be sacrificed but is replaced with

other dimensions of satisfaction in EPSS. For example, few customers were found fairly satisfied with the service performance due to the heating temperature not being as they expected. Demand response program reducing the heating temperature into minimum level to conserve energy and cut the operational cost. As a result, there are EPSS customers for whom satisfaction is reduced with regard to service performance but feel very satisfied with the economic performance and environmental performance. In other scenarios, some customers were found dissatisfied with POS, because it turned out they could not afford the AC they originally wanted to purchase and have to choose the affordable one, while EPSS service price does not suit their budget. We present satisfaction level for all scenarios in Appendix. Accordingly, the figure has provided supporting evidence for Hypothesis 2, where given the assumption of the proposed EPSS designs, EPSS customer satisfaction are in general equal to POS customers' satisfaction.

### 6.3. Company's Economic Benefit

Figure 14 exhibits cumulative economic benefit for the company from participating in the market. It attempts to evaluate whether proposed EPSS designs improve the company's economic benefit as stated in Hypothesis 3. Regarding profit generation, we previously clarified that the first and second service design generates profit from sales, while the third and fourth design generate profit from service sales and incentives from emission reduction programs. Eventually, the last four EPSS designs (the fifth, sixth, seventh and eighth design) generate profit from service sales and renewable energy sales.

In the figure, the economic benefit of POS Companies is presented as the cumulative benefit between the AC Company and Electricity Company. Its value is relatively higher compared to the average EPSS benefit. However, in reality POS Companies are separated firms, thus the benefit for each is not as higher as in the figure.

The designs with PV panel result in higher economic benefits compared to the non-PV design. Feed in Tariff from electricity sales double the company's benefit. The lowest economic benefit is from the first and second design, where revenue only comes from service sales. It seems that the more features offered to customer, the higher the economic benefits.

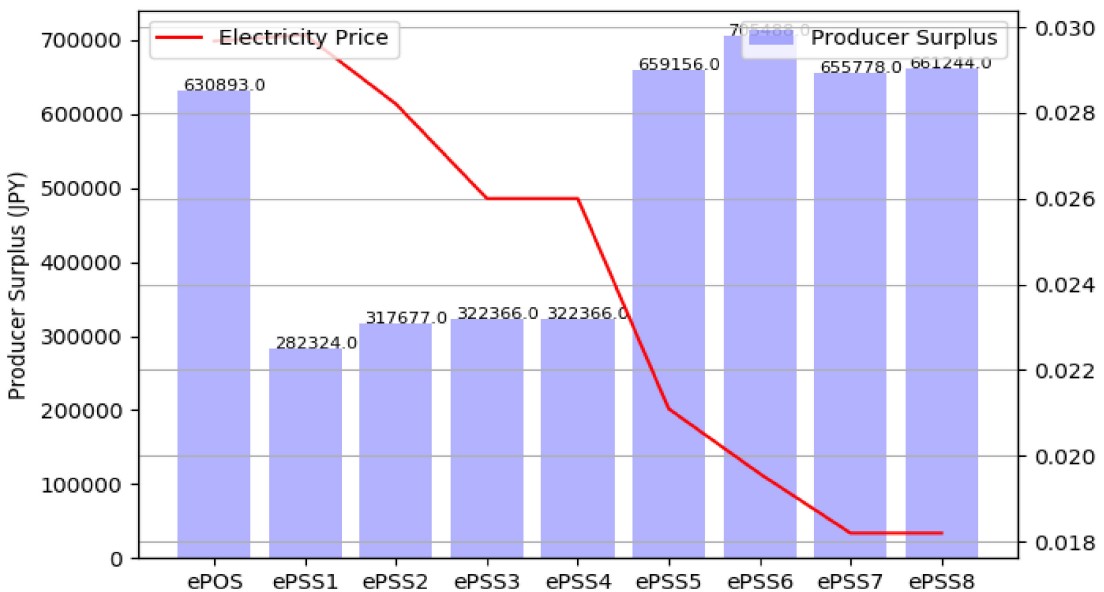

**Figure 14.** Producer Economic Benefit Comparison.

The red line shows electricity price average per KwH for each design. The trend of average electricity price in the market is lower for all EPSS scenario compare to POS. In this model, the demand response program and physical system optimization slightly reduce the average customer expenditure

for electricity per KwH in the market. Nonetheless, installed PV power generation significantly lowers the average expenditure for electricity per KwH in the market. In addition, note that EPSS gains more customer than POS. Accordingly, the result supports the evidence that EPSS in this model increase company's economic benefit due to service competitiveness and higher market share.

## 7. Discussion

Although this study strongly associates with the field of Product-Service System but the main contribution of this study is relevant to the energy system. It proposes a method to design and to evaluate new form of energy service and compare its economic performance with incumbent system. Commoditization has been said to reduce the product or service competitiveness in liberalized market. Likewise, electricity that has been commoditized throughout history faces difficulty to compete in a liberalized market. As a result, the energy service company's sustainability is endangered due to low profitability. In addition, from customer perspective, the existing service is indicated to have unclear benefit for the household.

Previous section displayed the result from simulating the proposed EPSS designs. It turns out that EPSS does not improve customers' economic benefit and that customers' expenditure to purchase EPSS service is higher than expenditure for POS service. We found that customers' benefit of EPSS does not come from the shifting ownership and not in form of cost efficiency, because basically ownership cost is transformed into another cost structure such as service cost.

Nevertheless, customers benefit is well defined in EPSS, in the way that the customer pays for service performance and levels instead of electricity. Therefore, households can monitor the quality of the service, which cannot be done with POS. As shown on Figure 15, EPSS and POS generate the same amount of heat to satisfy household demand, nonetheless, the amount of electricity required for heating varies depend on various factors. It implies that there's no direct correlation between the results of space heating temperature with electricity consumption. POS customer is charged for electricity supply, which does not reflect the quality of the space heating service. On the other hand, EPSS customers who pay for the service is allowed to focus on the quality of the output instead concerning the electricity supply service.

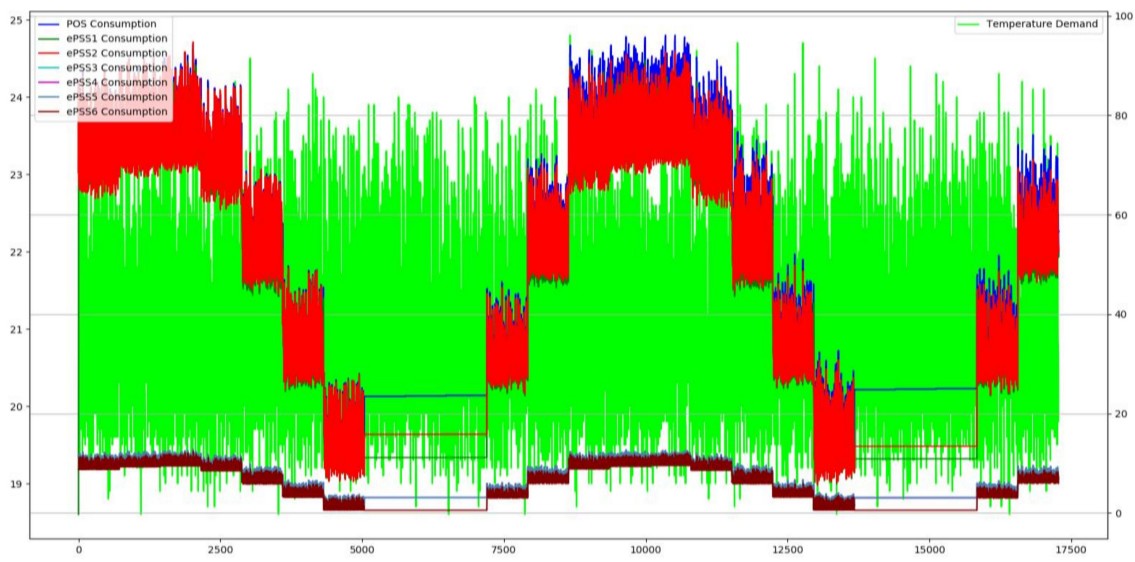

**Figure 15.** Comparison of Electricity Consumption for Heating.

Regarding companies' economic benefit, particularly in this model, EPSS designs generate benefit higher than individual POS companies. This is because we set EPSS service rate slightly higher than POS, yet still affordable for major customers in the market. Therefore, even with higher rates, customers that are rational decide to purchase EPSS services over POS products. This implies that

it is possible for EPSS to deliver competitive and affordable services just by integrating entities and technology from existing systems. Especially EPSS design that includes PV Panels and Feed-In-Tariff at current prices doubled the company's profit. Accordingly, we conclude that to create more benefit from EPSS, comprehensive consideration that covers electricity supply management, physical system optimization and consumption management is necessary to design the service.

## 8. Policy Implications

Our study offers several implications to improve the performance of energy sector and liberalized energy market. Japan Government through market liberalization aims to cut electricity price and expand business opportunities as well as provide more choice of electricity service for customers [63]. In addition, Japan energy mix target by 2030 is 22–24%, with solar energy share is the highest of renewable energy. Feed-in Tariff (FIT) Scheme has been the driving force in promoting renewable energy, which has significantly increased the capacity of facilities [63]. Unfortunately FIT surcharges for customer's bill becomes burden and has limited further growth.

Through EPSS, it is possible to maintain low electricity prices in a liberalized market. Using the same logic with households that do not need to purchase electricity if they have supplies from their own facility, we set electricity rates from PV panel for household consumption lower than the market price. In effect, the more EPSS services with the PV feature adopted, the higher possibility of holding down electricity rates in the market, while at the same time electricity retailers find other sources of revenue by providing competitive energy service for households.

Regarding energy mix target, from the simulation we found that EPSS can be used to facilitate the energy transition from brown energy into green energy without putting additional effort to influence the market preference to use green energy. With EPSS focusing on the quality of the end result of energy performance, energy retailers are encouraged to provide the most efficient energy source, which means renewable energy. The more customers adopt EPSS with the self-generated electricity feature, the higher renewable energy share in the total energy mix.

## 9. Conclusions

We have proposed a method to design and evaluate EPSS service and compare it with service under POS through Simulation-Based Design. Through the method, eight alternatives of EPSS service design are constructed from three service features comprises customer's demand management, operational system design and electricity supply management can be added and removed from the design. The result shows that EPSS design that includes operational system design and electricity supply management results in better benefit for all stakeholders. In this study, the customer's consumption management does not significantly affect the benefit due to insignificant amount of electricity consumption and low level of demand respond program on the simulation design.

At present method, alternative services are produced through brute-force approach, thus requires a very long time to run the simulation because it has to go through all possible combination and evaluate them. For further study, SBD for EPSS can be conducted by expanding the simulation method used for service evaluation. Instead of using brute-force approach, agent-based model can be implemented so that each stakeholder is provided with the ability to regulate him or herself and find which service alternative is optimum for each customer.

**Author Contributions:** W.K., T.T. and B.M. are all involved in conceptualizing the paper. W.K. designed the methodology, planned and carried out the simulation and conducted the formal analysis with input from all authors. W.K. wrote the original draft preparation and B.M. reviewed and edited the draft. T.T. and B.M., are both fully involved in supervising the research process.

**Funding:** This research received no external funding.

**Conflicts of Interest:** The authors declare no conflict of interest.

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
