# Peer review of "Designing and Evaluating Energy Product-Service Systems for Energy Sector (EPSS) in Liberalized Energy Market: A Case Study in Space Heating Services for Japan Household"

_challenges, doi:10.3390/challe10010018_

Round 1
Reviewer 1 Report
I have several suggestions for the authors.
First, the introduction sector needs to be strengthened, and claims such as "Unfortunately, competition between energy suppliers offering energy supply dominates the current liberalized market" or "Residential electricity demand is fundamentally derived from the demand for energy services, such transportation, space heating and lighting" should be supported by articles, reports or news.
Second, more papers can be included in the literature review, with identification of key contributions of this work. At present, the contributions are difficult to locate.
Third, in formulae, it is better to use symbols than words.
Fourth, assumptions of the model should be noted.
Fifth, policy implications are desirable.
Sixth, the quality of plots is in need of a significant improvement
Author Response
RESPONSES TO REVIEWER #1 COMMENTS:
We appreciate you taking the time to offer us your comments and insights related to the paper. We found your feedback very constructive. We tried to be responsive to your concerns. We hope you find these revisions rise to your expectations. Our responses follow (in italics)
1. COMMENT: The introduction sector needs to be strengthened, and claims such as "Unfortunately, competition between energy suppliers offering energy supply dominates the current liberalized market" or "Residential electricity demand is fundamentally derived from the demand for energy services, such transportation, space heating, and lighting" should be supported by articles, reports or news.
REPLY: Thank you for the valuable suggestion. The revised paper has been updated and references have been added to support the claims.
2. COMMENT: More papers can be included in the literature review, with identification of key contributions of this work. At present, the contributions are difficult to locate.
REPLY: We apologize for the vagueness. There’s a major change in literature review with more references added on it. We also have pointed out the contribution of the work, i.e. contributes to the method to design and evaluate EPSS economic benefit, in the first paragraph of section 7.
3. COMMENT: In formulae, it is better to use symbols than words.
REPLY: Thank you for pointing out this issue. As suggested, words for formulae have been replaced with symbols as shown in section 5.2
4. COMMENT: Assumptions of the model should be noted.
REPLY: Thank you for raising this important point. We have clarified the major assumptions in section 5.1
5. COMMENT: Policy implications are desirable.
REPLY: The policy implication has been stated in section 8.
6. COMMENT: The quality of plots is in need of a significant improvement
REPLY: Thank you very much for the suggestion. In the revised paper, the plot has undergone major changes. Hopefully, it is clear enough to convey the message of the paper.
To be more specified, we changed the order of Methodology and EPSS introduction. More references and figure in section 2 are added to introduce EPSS and provide a clear distinction between EPSS and POS, Typical PSS and Energy service.
For clarity, Methodology (section 3) is explained with figure. Section 4 and Section 5 (Case Study and Simulation Development) is provided with more explanation regarding customer, company and system behavior and property to clarify the system and simulation.
Reviewer 2 Report
This paper presents the new type of design and evaluate of energy systems. The authors show their own method EPSS and try to compare performance of EPSS to existing methods.
The paper shows how to prepare model, which parameters should be consider in it.
However, I have some doubts to the simulation part. How simulation is realized? There are no words about it. Which factors are taken to account, which algorithm? The results (Fig 4.) are sometimes similar and sometimes identical. In which grad is PV used as energy source? How to simulate persons behavior? Maybe some statistical data (averages) were used? How it was possible to get so detailed simulation like those on Fig. 9? It is not possible to evaluate simulations results without knowledge about simulation.
In conclusions it is written: “We have proposed eight alternatives of EPSS design for liberalized energy market as well as the simulation method to compare its performance with conventional energy service.” The eight alternatives are presented, but there is no simulation described. There are many results (Fig. 5 – Fig. 9) which looks good but cannot be verified without any information about simulation.
The paper looks good and interesting but without information about simulation it is not possible to accept it.
Author Response
RESPONSES TO REVIEWER #2 COMMENTS:
We would like to thank the reviewer for the thoughtful comments and constructive suggestions, which help to improve the quality of this manuscript. Our responses follow (in italics).
1. COMMENTS: This paper presents the new type of design and evaluation of energy systems. The authors show their own method EPSS and try to compare the performance of EPSS to existing methods.
The paper shows how to prepare model, which parameters should be considered in it.
However, I have some doubts to the simulation part. How simulation is realized? There are no words about it. Which factors are taken to account, which algorithm? The results (Fig 4.) are sometimes similar and sometimes identical. In which grad is PV used as energy source? How to simulate persons behavior? Maybe some statistical data (averages) were used? How it was possible to get so detailed simulation like those on Fig. 9 (Fig. 16 after revision) ? It is not possible to evaluate simulations results without knowledge about simulation.
In conclusions it is written: “We have proposed eight alternatives of EPSS design for liberalized energy market as well as the simulation method to compare its performance with conventional energy service.” The eight alternatives are presented, but there is no simulation described. There are many results (Fig. 5 – Fig. 9) which looks good but cannot be verified without any information about simulation.
The paper looks good and interesting but without information about simulation it is not possible to accept it.
REPLY: Thank you for raising this important issue. In the revision, we provided more explanation about simulation, particularly in Section 3. Methodology. We have explicitly mentioned the steps that have been conducted for the research including the tools and the expected result from each step. Furthermore, detail description of the system and assumptions of customer and company’s properties and behavior are provided in section 4 and 5. Hopefully, it can clarify the simulation development to get all the figures in section 6.
Round 2
Reviewer 1 Report
Since most of the problems have been addressed, I recommend the paper to be published.